# Optimising Web-Based Computer-Tailored Physical Activity Interventions for Prostate Cancer Survivors: A Randomised Controlled Trial Examining the Impact of Website Architecture on User Engagement

**DOI:** 10.3390/ijerph17217920

**Published:** 2020-10-28

**Authors:** Amy Finlay, Holly Evans, Andrew Vincent, Gary Wittert, Corneel Vandelanotte, Camille E Short

**Affiliations:** 1The Freemasons Foundation Centre for Men’s Health, School of Medicine, University of Adelaide, Adelaide 5000, SA, Australia; amymfinlay@gmail.com (A.F.); holly.evans@adelaide.edu.au (H.E.); andrew.vincent@adelaide.edu.au (A.V.); gary.wittert@adelaide.edu.au (G.W.); 2Physical Activity Research Group, Appleton Institute, Central Queensland University, Rockhampton 4701, QLD, Australia; c.vandelanotte@cqu.edu.au; 3The Melbourne School of Psychological Sciences and Melbourne School of Health Science (Jointly Appointed), The University of Melbourne, Parkville 3010, VIC, Australia

**Keywords:** computer tailoring, e-health, cancer survivorship, user engagement, physical activity, behaviour change, website architecture

## Abstract

Background: Web-based computer-tailored interventions can assist prostate cancer survivors to become more physically active by providing personally relevant behaviour change support. This study aimed to explore how changing the website architecture (free choice vs. tunnelled) impacted engagement within a physical activity computer-tailored intervention targeting prostate cancer survivors. Methods: On a 2:2:1 ratio, 71 Australian prostate cancer survivors with local or locally advanced disease (mean age: 66.6 years ± 9.66) were randomised into either a free-choice (N = 27), tunnelled (N = 27) or minimal intervention control arm (N =17). The primary outcome was differences in usage of the physical activity self-monitoring and feedback modules between the two intervention arms. Differences in usage of other website components between the two intervention groups were explored as secondary outcomes. Further, secondary outcomes involving comparisons between all study groups (including the control) included usability, personal relevance, and behaviour change. Results: The average number of physical activity self-monitoring and feedback modules accessed was higher in the tunnelled arm (M 2.6 SD 1.3) compared to the free-choice arm (M 1.5 SD 1.4), *p* = 0.01. However, free-choice participants were significantly more likely to have engaged with the social support (*p* = 0.008) and habit formation (*p* = 0.003) ‘once-off’ modules compared to the standard tunnelled arm. There were no other between-group differences found for any other study outcomes. Conclusion: This study indicated that website architecture influences behavioural engagement. Further research is needed to examine the impact of differential usage on mechanisms of action and behaviour change.

## 1. Introduction

Physical activity is recommended as an important part of prostate cancer survivorship. This is due to the many physiological (e.g., increased bone loading, muscle endurance, and weight loss) and psychological benefits that can improve men’s quality of life during and post-cancer treatment [1,2]. However, the vast majority of prostate cancer survivors are not participating in sufficient physical activity to obtain health benefits [3]. In order to support prostate cancer survivors to increase their level of physical activity and improve their quality of life, accessible and relevant support is needed.

Computer-tailored interventions are a promising method to provide online behavioural support for prostate cancer survivors [4,5,6,7]. Computer tailoring is a feasible, acceptable and efficacious method to deliver personalised and iterative behaviour change support en masse [8,9] and has been used in an oncology setting in recent years [10,11,12,13]. In brief, a computer-tailored program algorithmically maps a unique combination of messages to an individual. The personalised messages are based on pre-measured behavioural, psychological, medical or demographical characteristics [14].

Despite the interest and gains made in computer tailoring, there is still room for improvement. Many studies report issues with non-usage attrition that compromises a program’s efficacy [15]. That is, exposure to persuasive behaviour change techniques presented in the later parts of a program is often compromised due to participant disengagement. Online engagement is broadly defined as the extent of usage and the subjective user experience that is characterised by attention, interest and affect [16]. It is thought to impact program effectiveness by influencing the depth of involvement with the behaviour change process (e.g., effort and attention towards set goals) [16,17]. Therefore, exploration and experimentation are required to understand what aspects of programs could potentially influence online engagement.

One under studied area is the impact of program architecture of computer-tailored websites. Traditional computer-tailored web-based interventions use ‘tunnelling’ techniques [14,18]. Tunnelled programs provide small packages of information in a set order delivered over time to avoid overwhelming the participant and to guide them through the behaviour change process. While reducing burden and providing guidance are clear advantages, a potential limitation of this method is that it reduces user autonomy. Greater autonomy is associated with increased intrinsic motivation to perform a behaviour, and greater behavioural maintenance in turn [19,20]. As such, designing autonomy supportive interventions may result in greater likelihood of on-going engagement [21]. Autonomy may be especially important for prostate cancer survivors, as autonomy and control can by threatened by a prostate cancer diagnosis [22]. However, the impact of website architecture has not been explored in a controlled setting among prostate cancer survivors.

There has been limited experimentation with computer-tailored intervention website architecture in general. Short et al. (2017) examined the impact of tunnelled modules delivered via different schedules among breast cancer survivors and found a trade-off effect between completion of modules (favouring a weekly group) versus acceptability and actual behaviour change (favouring a monthly group) [23]. A systematic review found only three website architecture experimentation studies [24]. Two of the studies compared a tunnelled tailored condition to a non-tailored control [25,26]. The tailored versions were superior, which is unsurprising as tailored interventions are generally more effective than non-tailored interventions [27]. The final study in the review compared a tunnelled tailored program aiming to influence knowledge of hepatitis, to a tunnelled tailored version where participants could skip website pages and non-website control [28]. Those in the skipping-pages group reported higher perceived acceptability of the website, though the standard tunnel had greater information retainment post-intervention. The extent to which information retention relates to behaviour change is unclear, though these outcomes suggest a trade-off effect. Skipping pages within a linear tunnelled structure also only provides partial autonomy to the users. The impact of a completely autonomous structure compared to a tunnelled structure has yet to be examined. It may be that autonomy supportive interventions allow people to self-tailor, and that this may increase perceived intervention relevance. This could have positive impacts on both engagement and efficacy, depending on individual’s ability to self-tailor content to meet their needs.

A comparison of a computer-tailored behaviour change program that differs not in content but *only* in website architecture is needed. The primary aim of this randomised controlled trial was therefore to explore online engagement differences between a standard ‘tunnelled’ and a free-choice (autonomy supportive) behaviour change intervention among prostate cancer survivors. The secondary aim was to collect data on acceptability, website relevance and preliminary efficacy of the two interventions in terms of physical activity behaviour change compared to a control group that received standard online Australian prostate cancer resources.

## 2. Materials and Methods

### 2.1. Study Design

This trial was a parallel, three-arm randomised controlled trial with two computer-tailored intervention arms (tunnelled and free choice) and a non-tailored control. Participants were randomised on a 2:2:1 ratio by a computer-generated algorithm embedded within the website. The study protocol was registered prospectively with the Australian New Zealand Clinical Trials Registry (identifier ACTRN 12618000-808246) and the University of Adelaide Human Research Ethics Committee (H-2017-101, approved 27 June 2017) granted ethics. This study adheres to the Consolidation Standards of Reporting Clinical Trials (CONSORT) [29].

### 2.2. Procedure

Potential participants were informed that the purpose of this study was to evaluate a computer-tailored physical activity intervention, and that they would be randomised to receive either the tailored intervention or the control intervention. Participants were not informed that there were two tailored website arms and were thus blinded to the true aim of this study. Interested participants were directed to the study website for eligibility screening, which included a safety check for physical activity participation (PAR-Q, [30]). Eligible participants were able to register for the study online, and were automatically randomised and given access to the study website after providing consent. Participants first needed to complete the baseline survey via the website before they were able to access the intervention group home page. The intervention period was for four weeks. This intervention time frame was considered adequate given that other web-based modular intervention studies have demonstrated significant non-usage attrition within one to four weeks [23,31]. The post-intervention survey was completed via the study website at the end of the four-week intervention. Participants received a personal reminder email and a follow-up call from the lead researcher (AF) if investigators did not receive the final research survey after one week.

### 2.3. Selection Criteria

Participants were required to be living in Australia, over the age of 18 years, diagnosed (any time) with localised or locally advanced prostate cancer (stage 1–3) and have completed primary treatment (any time), be in remission, not already meeting the aerobic and resistance training components of the oncology physical activity guidelines (meaning they could be meeting one component and be eligible) [32,33], be able to read and write in English, have access to a computer and the Internet, and have no medical contraindications to exercise. Exclusion criteria included prostate cancer survivors who were on active surveillance, or with terminal and/or metastatic disease, those who had medical contraindications to exercise, and did not have a doctor’s permission to participate (where deemed necessary by the screening tool).

### 2.4. Recruitment

Recruitment for this study was multifaceted. Existing networks and infrastructure were utilised, including the University of Adelaide’s Men’s Health Register and the Pathfinder Prostate Cancer Research Register. This study was advertised directly to health professionals through professional networks (e.g., national urological nurses’ newsletter), known contacts and conference presentations (e.g., urologists and nurses), and directly to consumers by researching out to support and advocacy groups, and via Facebook advertising.

### 2.5. The Interventions

The interventions were marketed as Prostate Cancer Health and Fitness (PCHF) online (www.pchf.net.au) which promoted the uptake of both aerobic and strength-based activity to Australian prostate cancer survivors. Both intervention arms contained the same computer-tailored feedback and differed *only* in the website architecture (see Figure 1). The intervention content targeted physical activity determinants outlined by Social Cognitive Theory [34] and habit theory [35], and applied previous theory mapped behaviour change techniques [36,37] as per Appendix A. The messages used autonomy supportive language, in line with promoting intrinsic motivation (relatedness, competence and autonomy), drawing from Self-Determination Theory [19]. For example, using relatedness statements such as “you’re not alone” or “many people”, or promoting autonomy by “what might work for you?” The content was also informed by a two-part qualitative study conducted by the research team (N = 16; to be published elsewhere). The findings of the first study suggested that messages should tailor to baseline physical functioning, goal setting (formal or informal), take into account different social support preferences, and provide information in short, simple, easy to understand messages written in causal language. Interviews indicated that most participants value their health but find it hard to stay motivated due to issues such as pain, injury, finding the time, general motivation and weather. Addressing these factors was deemed important to increase the relevance of physical activity messages targeting prostate cancer survivors. A behavioural scientist (CES) and an exercise physiologist (HE) screened the messages for accuracy, flow and safety.

#### 2.5.1. Intervention Content

There were two forms of content contained within PCHF. The first was considered ‘once-off’ content. That is, content that does not require on-going input, such as describing the benefits of exercise. The intervention contained four ‘once-off’ advice modules; see Appendix A, which were theoretically mapped to Social Cognitive Theory. These components were known as “getting started”, “goals and barriers”, “lone ranger or socialite?—Exercising with others” and “making long-term changes”.

In brief, the “getting started” modules provided an overview of oncology physical activity guidelines, the benefits of physical activity, and provided tailored information based on age and comorbidity functional impairment status. The “goals and barriers” module described basic and advanced goal-setting approaches, as well as tailored advice for addressing personal barriers (e.g., time, weather, low motivation, and incontinence). The “lone ranger or socialite?—Exercising with others” module contained advice relating to social support, tailored to participant exercise preferences. Finally, the “making long-term changes” advice module provided tailored information regarding habit formation, relapse prevention, motivation, and general links to further support.

The second form of intervention content, the physical activity logs, were designed to take inputs from users across time into account and provide iterative feedback. There were four possible logs to complete, each assessing activity over the last week. These modules utilised goal setting and self-monitoring based on Social Cognitive Theory. Each week (except the first), the module provided feedback regarding the participant’s previous week’s goal. The module also prompted participants to think about the next week’s goal. Graphical representation of participant’s self-reported moderate–vigorous minutes and resistance training was provided. Each week’s graph had the previous data for participants to look at the patterns over time.

#### 2.5.2. Additional Website Features

Both intervention arms contained additional features including an “Ask an Expert” email function that allowed users to submit a question to an accredited exercise physiologist. The answer would be sent directly via text and video from exercise physiologist (HE), and the answer was also placed upon the website for all participants to see. There was also a library section that contained hyperlinks to pre-recorded videos demonstrating resistance training exercises (with and without a resistance band).

To support participants with additional information needs, extra in-depth articles on prostate cancer and health were provided in the library. This included links to scientific articles on prostate cancer, as well as general prostate cancer survivorship topics, such as sexual wellbeing, diet and exercise (see Appendix A). Participants were sent automatic emails twice a week to remind them to log into the website. For screen shots of the website home pages; see Figure 2 and Figure 3.

### 2.6. Study Arms

#### 2.6.1. Standard Tunneled Intervention

Participants randomised into the standard tunneled arm received a single weekly module that combined ‘once-off’ advice and a physical activity log. As is common in computer-tailored interventions [18], the health advice modules were ‘drip-fed’ to users in a logical order based on the proposed process of behaviour change (see Appendix A). The participant could only access the next combined module after seven days had elapsed from completion of the prior module (see Figure 1 and Figure 2).

#### 2.6.2. Free-Choice Intervention

Free-choice participants had the same content as the standard tunneled condition (see Figure 1 and Figure 3). However, the ‘once-off’ tailored advice modules were presented as stand-alone modules and could be accessed at any time and in any order. The physical activity log modules were also presented as stand-alone modules, with the relevant log for that week presented on the home page.

#### 2.6.3. Non-Tailored Minimal Control

Those randomised to the control arm had access to a home page that contained static information about the oncology guidelines and links to high quality, freely available Australian prostate cancer websites akin to usual care. For example, participants received a link to exercise recommendations from the Australian Cancer Council; see https://www.cancersa.org.au/information/a-z-index/exercise-for-people-living-with-cancer. However, this information was not tailored, and the control did not have access to the library or the ‘Ask an Expert’ function. After completing the final survey, participants randomised into the control arm were offered a chance to use either version of the intervention (based on their preference reported at baseline).

### 2.7. Measures

#### 2.7.1. Participant Characteristics

Demographics collected at baseline via the self-report survey included age; education; marital status; employment status; postcode, recoded into remoteness levels using an online tool based on the Accessibility Remoteness Index of Australia [38]; cancer stage; cancer treatment(s); time since diagnosis; weight and height (to calculate BMI), and type of comorbidities (recoded into number). Intervention architecture preference was also collected via the self-report survey at baseline. Participants were also asked whether they would ‘hypothetically’ prefer a free-choice version (“one that allows you access to any of the topics whenever you want and provides feedback as you go”) or a tunnelled version of the intervention (“one that guides you through topics step-by-step and provides feedback as you go over a couple of weeks”). Google Analytics were used to assess the number of individuals who accessed the study website (including during recruitment).

#### 2.7.2. Primary Outcome

Self-monitoring is considered one of the most efficacious behaviour change techniques for promoting physical activity behaviour change [39,40]. Therefore, the primary outcome was the difference in completion rates of the four physical activity logs between the participants in the free-choice and tunnelled intervention arms, determined by assessing differences in total number of physical activity logs completed at follow up (possible scores ranging from 0 to 4). All website usage data were collected automatically via the study website.

#### 2.7.3. Secondary Outcomes

##### Other Website Usage

In addition to the total number of physical activity logs completed, the number of users accessing the stand-alone health advice module content, and minutes spent accessing the library page were also assessed and compared between the two intervention groups.

##### Physical Activity

Participants’ minutes per week of moderate–vigorous activity (MVPA) was measured at baseline and at the immediate intervention follow up using an adapted version of validated self-report Godin Leisure-Time Exercise Questionnaire (GLTEQ) [41,42]. The tool was adapted by asking participants to estimate the number of minutes of activity participated in for each session, rather than the number of sessions only. Participants were asked to report based on the average weekly amount over the last month. To determine moderate–vigorous physical activity minutes per week, individual scores for the moderate–vigorous aerobic activity were added together and vigorous activity was multiplied by two (to account for the additional benefits associated with vigorous physical activity). Three resistance training questions for the number of sessions, exercises and repetitions were also asked, as per [42]. A total resistance training score was calculated by multiplying the number of sessions by the number of exercises reported, where higher scores indicated greater participation. The proportion of participants meeting the oncology physical activity guidelines (>150 min of MVPA+ ≥ 2 resistance sessions per week (4)) was also computed from these variables.

##### User Perceptions

Multiple measures were collected in the post-intervention follow up to examine user perceptions of the intervention in terms of their subjective experience and overall satisfaction levels.

The 12-item e-health engagement scale [43] was used to measure participant’s subjective experience of the intervention relating to engagement. This scale asked participants “*to what extent did you find the program …*” on a series of characteristics such as “cool”, “trustworthy” or “stimulating” with a 5-point Likert scale from strongly disagree to strongly agree. The average score is taken across the 12 items (max 5), where higher scores indicate a more positive subjective user experience.

Website usability was measured through the 10-item System Usability Scale (SUS), with a score above 68 indicating “above average usability” [44].

The perceived relevance of the intervention content was assessed using 3 items adapted from a previous computer-tailored intervention for breast cancer survivors [23]. Participants were asked whether they thought the messages in the program were (a) very relevant to me, (b) very applicable to me, and (c) the messages felt like they had someone like me in mind on a 5-point Likert scale from strongly disagree to strongly agree. An average of the items was taken with higher scores (max 5) indicating higher perceived relevance.

Open-ended questions were asked to participants in order to gain general qualitative feedback. This included assessing participant’s opinion of the exercise physiologist “Ask an Expert” feature and why they did or did not use this section. Participants were also asked to give feedback regarding the pros and cons of the website and suggestions for improvement.

### 2.8. Data Analysis

All statistical analyses were performed using SPSS version 25 and Stata version 15.1. Descriptive statistics were used to describe the trial population. Differential dropout between groups was explored using chi-square tests. Non-adjusted and adjusted models were calculated for all study outcomes, with non-adjusted models considered the primary analysis. The covariates included in the adjusted models included age, location, education, time-since treatment and baseline physical activity levels. These were selected a priori. Differences in the number of modules completed between free-choice and tunnelled intervention arms (the primary outcome) were compared using proportional odds regression. Two-group comparisons between free-choice and tunnelled intervention arms for continuous variables were conducted using *t* tests in unadjusted models and ANCOVA in adjusted models. Chi-squared/Fisher’s exact tests were used for the two-group comparisons of categorical data (unadjusted), with logistic regression used for adjusted models. Three-group comparisons (free choice, tunnel and control group) examining differences in user perceptions and continuous physical activity scores were conducted using ANOVA (unadjusted) and ANCOVA (adjusted). Logistic regression was used to examine differences between groups in meeting the physical activity guidelines. All statistical analyses were conducted using all observed data (i.e., complete case analysis). A sensitivity analysis was conducted to explore the impact of missing physical activity data by imputing baseline levels. This resulted in no differences in interpretation of data, and so complete case data are reported only.

Qualitative data were examined for common themes as well as for unique points of view for intervention improvement.

#### Sample Size Calculation

The primary end point was the total number of completed weekly physical activity log modules. The primary analysis was a two-group comparison between the free-choice and standard tunnelled arms after four weeks. With an assumed constant weekly intervention attrition rate of 40% in the tunnelled arm [23,31,45] and assuming the number of weeks completed is Poisson distributed (noting that the square root of a Poisson distribution is approximately normally distributed with variance 0.25), randomising 112 individuals equally between groups provided 75% power to detect a mean difference of 0.25 (i.e., change in attrition from 40% in ‘fixed’ to 15% in ‘autonomous’ using square root transform) in a two-group t test (two-sided alpha = 0.05). Randomising one-fifth to a control group (randomisation ratio 2:2:1) results in a total sample size of N = 135, in block sizes of six. This would create goal recruitment for N = 54 in each of the experimental arms and 27 in the control. While a control group was not required for the primary outcome, the presence of a control group was required to assist interpreting results relating to acceptability and efficacy of the intervention for the secondary outcome.

## 3. Results

### 3.1. Participant Flow

The participant flow, including study retention, is represented in Figure 4. There were 411 individuals who clicked on the online screening tool between 8 August 2018 and 15 March 2019. However, 255 individuals exited the webpage. There were 156 screening tool completions, of which 50% of individuals were eligible, registered, and were randomised to a trial arm (N = 78). The majority of participants were recruited through social media (34.6%), support groups (19.2%) and a national prostate cancer survivors research registry (19.2%). After randomisation, 71 participants completed the baseline questionnaire and were provided access to their website condition (N = 27 free-choice group, N = 27 standard tunnel, and N = 17 control). Of these, 70% of participants (N = 50 total; N = 16/27 free choice, N = 20/27 standard tunnel and N = 14/17 control) completed the post-intervention follow-up survey. The proportion of participants completing the post-intervention survey was lowest in the free-choice group (59%), followed by participants in the tunnelled group (74%), with control group participants having the highest completion rate (82%). Differences in dropout were not statistically significant (*p* = 0.23). As website usage data were collected automatically, usage data for 100% of participants who accessed their website condition (N = 71) are available (i.e., 91% of those randomised).

### 3.2. Participant Characteristics

Participant demographic and health characteristics are presented in Table 1 (N = 71). In general, participants were well educated (81% educated beyond high school), partnered/married (85%), retired (53.5%) or working full time (22.5%), from metropolitan (45%) or inner regional centres (27.3%) and represented all states and territories of Australia. Participant health characteristics were more varied, with a range of cancer stages, treatments, co-morbidities and body mass indexes reported. The average time since treatment was three years, with the most common treatment reported as surgery (79%), followed by radiotherapy (35%). On average, participants reported high aerobic activity levels at baseline (Mean MVPA = 284.37 min/wk, SD 211.95), and low–moderate resistance training levels (M = 1.6 sessions per week, SD 28.3; see Table 2)).

Participants who completed the post-intervention follow-up survey were more likely to have higher levels of education compared to non-completers (*p* = 0.03). Completers were also likely to have completed their treatment more recently, and were more active at baseline. However, these were not significantly different (Ps > 0.05).

When asked about preferred intervention architecture at baseline, 62% of participants selected the “free-choice” option and 38% selected the “tunnelled” option.

### 3.3. Primary Outcome

From a maximum score of 4, the average number of physical activity logs completed was higher in the tunnelled arm (M 2.6 SD 1.3) compared to the free-choice arm (M 1.5 SD 1.4). Though both groups’ engagement reduced over time (see Figure 5). According to the unadjusted proportional odds regression model, a one-unit increase in group (i.e., going from the free-choice arm to the tunnelled arm) was associated with a 1.85 unit increase in the log odds of having a higher level of physical activity log competition (95% CI 0.43, 2.45; *p* = < 0.001; adjusted model *b* 1.85, CI 0.69, 3.01, *p* = 0.002).

### 3.4. Secondary Outcomes

#### 3.4.1. Other Website Usage

As both the “once-off health advice” modules and the physical activity logs were embedded in the same larger component in the tunnelled arm, the overall health advice module engagement rates (i.e., 96% 74%, 52% and 41%) in the tunnelled arm are the same as the engagement rates of the physical activity logs (See Figure 5). Tunnelled participants could only access the final two “once-off” components (i.e., social support and habit formation) by staying in the program into weeks 3 and 4. In contrast, the free-choice participants could engage with the “once-off health advice” modules at any time. There were no significant differences in proportions of participants between groups accessing the “getting started” and “goals and barriers” modules (all Ps > 0.05). However, significant differences were observed between groups for accessing the social support (*p* = 0.018, adjusted *p* = 0.01) and habit formation modules (*p* = 0.002, adjusted 0.004), with a greater proportion of participants in the free-choice group accessing this content compared to the tunnelled group.

The library function was used by 29.6% of participants, with an average usage of 3.1 min (SD 5.3 min, range 1 s to 20.8 min). There were no statistically significant differences in minutes of use between intervention arms (All Ps > 0.05).

The “Ask an Expert” feature was used by one participant from the tunnelled arm. No participants in the free-choice arm used this feature.

#### 3.4.2. Physical Activity

There were no statistically significant differences between groups in moderate–vigorous aerobic activity or resistance training scores (number of sessions * number of exercises) for unadjusted (expect for baseline activity levels) or adjusted models; see Table 2. There was an increase across groups in the percentage of participants meeting the guidelines relative to baseline scores (free choice +25%; tunnelled +20%; control +36%). However, there were no significant between-group differences (All Ps > 0.05). Geographical location was a significant predictor in adjusted models for both resistance training score (Table 2) and meeting the physical activity guidelines (OR 3.79, 95% CI 1.07, 13.44), with living outside of a major city area positively associated with physical activity scores at follow up compared to living within a major city area (adjusting for baseline physical activity, group, age, education and time since treatment).

#### 3.4.3. User Perceptions

Overall, the self-reported engagement, usability and relevance scores were low to moderate across groups, with no significant differences between arms in adjusted or unadjusted models (>0.05); see Table 3. The average usability score for participants allocated to the tunnelled arm (M = 67.4) was close to the cut off for ‘above average usability’ (>68), and was the highest of all study groups. However, this was not significant at the 0.05 level (*p* = 0.06).

#### 3.4.4. Qualitative Feedback

There were 29 participants who provided written feedback in the open-ended text box about pros and cons of the website and recommendations for improvement (N = 8 free choice, N = 12 tunnelled, N = 9 control). The qualitative feedback from those in the intervention groups did not tend to differ between groups and was moderately positive in tone. Three participants indicated that the program concept was valued, though the execution needs work. For example, intervention comments included “a fine effort and well worth continuing”, “a good start” and that “the programme would be fantastic for anyone who doesn’t know where to start or has no backup”. Two participants specifically mentioned the hyperlinks in the library and considered them important (hyperlinks related to sexual health information and exercise videos). Issues of content relevance were noted, for example, comments from individuals included “better feedback is needed”, “I did not find many of the activities applicable to me or my lifestyle”, and “I need a program that works on my sexual fitness not my physical fitness”. Potential improvements suggested by participants included providing a higher variety of resistance exercises for different levels of fitness, tailoring more precisely for age and providing “more details on the various cancer treatments and possible effects”. Suggestions also included using the “best bits” of the website into a mobile app or combining the program with a face-to-face group. Of the 27 participants in the intervention arms who gave feedback on the Ask an Expert feature, approximately half felt that it was unnecessary for them. Other reasons for non-use included a lack of time and not noticing the feature on the website.

For those in the control arm, four control participants specifically said that they continued with their “own routine”. Two participants reported that the control version of the website was either “underwhelming”, or “I did not feel that I was participating in an exercise program at all.”

## 4. Discussion

The results of this study indicate that website architecture of Prostate Cancer Health and Fitness online (PCHF) somewhat impacted behavioural engagement between the intervention arms. Findings for the primary outcome (physical activity log engagement) suggest that the standard tunnelled condition was relatively more successful in engagement and potentially more usable than the free-choice arm. However, both intervention arms accessed approximately 60% of the total available intervention content (four physical activity logs + four ‘once-off’ advice modules), yet each arm accessed different aspects of the website. A greater proportion of participants in the free-choice arm were likely to be exposed to the ‘social support’ and ‘habit formation’ content than those in the tunnelled arm. However, this exposure was at the expense of self-monitoring and weekly goal setting over time, which targeted self-efficacy, and is an aspect considered to be important for behaviour change [34,36]. The tunnelled group, in contrast, had the same issue but in reverse. The impact of this trade off on the user experience and behaviour change outcomes is unclear in the present study, with no or very minor between-group differences. It does seem, however, that to encourage self-monitoring, a tunnelled approach where the user is guided through the intervention is more desirable.

Trade-off effects have been noted in previous website architectural experiments [23,28]. An RCT by Crutzen et al. [28] investigating the impact of being able to skip pages in a tunnelled website versus having no control to skip pages showed that greater control led to greater perceived website efficiency, but less time spent on the website, fewer website pages visited and lower (hepatitis) knowledge gained. They concluded that the findings demonstrate increased website usage in the standard tunnelled arm and that visitors should therefore be carefully guided through future intervention websites. A three-arm RCT by Short et al. [23] investigated the impact of manipulating the delivery schedules of three tunnelled modules (monthly, weekly versus single module control) focusing on physical activity promotion among breast cancer survivors. As well as receiving the tailored module, all participants were provided with a weekly action planning tool and were instructed to engage with the intervention website for 12 weeks. The results indicated that those in the monthly schedule arm reported higher website acceptability and greater resistance training compared to the control, and also completed a greater number of action plans compared to the weekly module group. However, the completion of at least two tunnelled modules was higher in the weekly module group compared to the monthly module group, presumably due to non-usage attrition over the 12 weeks [33]. The findings by Short et al. highlight that usage or completion of tunnelled modules alone should not be the primary area of interest to those wishing to optimise efficacy of behaviour change websites. Consideration needs to be given to all website features and how they contribute to (a) user burden and (b) impacting on important mechanisms of behaviour change. In this case, how module engagement relates to changes in mechanisms of action (i.e., changes in social support, changes in self-efficacy, and habits), and therefore efficacy.

Of note, as typically reported in other computer-tailoring studies, many participants in PCHF did not return to the website to track their behaviour using the physical activity logs, regardless of architecture. This might be best addressed by removing the requirement to manually log data and utilising sensors instead. Previous computer-tailored research has already shown that both attrition and efficacy are improved when sensors are used to collect physical activity information for tailoring in place of manually logging physically activity [46]. In fact, research into wearable trackers in cancer populations is gaining traction [5,47], and has been found acceptable [48] and valid [49] in prostate cancer populations. However, simply giving prostate cancer survivors a wearable tracker to use would unlikely to be a ‘silver bullet’ to meet physical activity and behaviour change needs [50]. Current wearable trackers are disease non-specific, and therefore adaption to allow for cancer-specific exercise prescription and behaviour change needs may be required [6,51]; for example, step count goals that are adjusted based on treatment-related side effects and behaviour change support that acknowledges unique barriers relating to cancer symptoms (e.g., incontinence and fatigue). It is recommended that further research explore the use of wearable moment-to-moment behavioural tracking in combination with computer-tailored feedback among cancer populations. With automated tracking, the free-choice model may be improved. Given that most participants at least like the idea of having choice and control, designs that can satisfy this desire whilst also providing appropriate guidance and exposure to core behaviour change techniques may result in both increased adoption and enhanced behaviour change outcomes.

Another noteworthy finding in this study is that relevance scores did not differ between groups. Given that the free-choice arm allows people to “self-tailor” according to their interests or needs, one might expect perceived relevance to be highest in this group. However, there were no differences in the perceived relevance of website content across groups, including with the control group that was only provided links to targeted (but not tailored) information. Overall, perceived relevance was moderate across groups, and suggests that in this instance the tailoring did not increase perceived relevance compared to targeted information about prostate cancer survivorship. Previous studies in cancer populations have shown that tailored information is perceived as more relevant than targeted information, though achieving high perceived relevance scores across all participants remains a challenge [13,45,52]. The qualitative feedback suggests that tailoring more precisely for demographic and health information may be needed to boost tailoring effects. In the current study, these factors were included as tailoring variables, but tailoring was predominantly based on determinants of exercise (e.g., self-efficacy) and physical activity behaviour. Previous computer-tailoring studies suggest that taking functional impairment into account, in terms of age, comorbidities and treatment toxicities is important for content relevance [53], and this was also reflected in our formative research. Another likely contributing factor, somewhat on the other end of the spectrum, is the high levels of baseline aerobic physical activity observed in this study. Most participants entered this study based on a lack of resistance training and were already highly aerobically active. This may have reduced the perceived relevance of much of the tailoring, given that the behaviour change determinants targeted (e.g., self-efficacy, social support and habit formation) may already have been largely addressed. Participant’s interest is resistance training is reflected in the qualitative feedback, with participants noting appreciation of the resistance training videos included in the library, and a desire for a more varied and tailored resistance training program to follow. In order to be able to observe any added benefits of self-selecting tailored modules rather than being guided through them, these limitations in the tailored content need to be addressed.

### Strengths and Limitations

This study was supported by a strong methodology using a randomised controlled trial design. Participants were blinded to the main outcome of this study, and the objective website data for the primary outcome were available for 100% of all individuals who received their allocated intervention. The intervention design was evidence-based, rigorous, and was grounded in behaviour change theory. Furthermore, unlike other website architecture experiments [24], computer tailoring occurred in both intervention arms. This placed the study emphasis on the website architecture, rather than the tailored content itself. However, there are study limitations to consider.

Over half of the participants who clicked on the screening tool did not go on to complete the screening (255/411). It is not known how many of these may have been eligible to participate. Of those participants who did complete the screening eligibility tool, 49% enrolled in this study, which is higher than [5,31] or comparable to [7] other studies in this field. Once enrolled, PCHF had a study retention rate of 64% of all participants randomised (N = 50/78), and 70% of those who received their allocated intervention (N = 50/71). This is similar to Kenfield’s recent behaviour change prostate cancer study [5] but is lower than other prostate cancer online studies [4,7]. Given the relatively short duration of the current study, and that dropout tends to increase overtime, a higher retention rate could be expected in our study. Further, even though the recruitment period was doubled in time, this study did not reach the recruitment target, similar to issues noted in other men’s health research [54,55]. This indicates a major feasibility issue, and limits the ability to discern differences, particularly in secondary outcomes.

Although this study was primarily designed to investigate differences in website usage using objective measures, self-report was relied upon for assessing changes in physical activity. Objective measures, such as accelerometry and strength tests (as a proxy for resistance training), may have allowed us to explore the impact of group allocation with more precision [56]. Large standard deviations were observed for physical activity in the current study, which is typical of self-report, and negatively impacted on power. Objective measures, in combination with stricter inclusion criteria and perhaps an active control condition (e.g., nutrition intervention) may have allowed us to better explore the relationship between website architecture and behaviour change. Qualitative feedback suggests that in the absence of an intervention, control group participants found other ways to maintain or increase exercise. This study also only focused on short-term changes in behaviour change and did not include a longer-term follow up.

Nevertheless, this study was able to gain a broad national sample with participants from all Australian states and territories. Despite not meeting the recruitment target, this study was still able to identify significant differences in the main outcome. Future prostate cancer studies could utilise additional recruitment avenues, such as the use of cancer registries to obtain more representative samples.

## 5. Conclusions

The website architecture of computer-tailored interventions is likely to impact online engagement. This study provides further support that a tunnelled approach may be best suited to encourage self-monitoring of physical activity. However, there are important trade-offs to consider that may impact on intervention efficacy among men with prostate cancer. Further research is required to explore links between website architecture, engagement and efficacy. Exploration of wearable trackers alongside different website architecture within computer-tailored interventions is also recommended.

## Figures and Tables

**Figure 1 ijerph-17-07920-f001:**
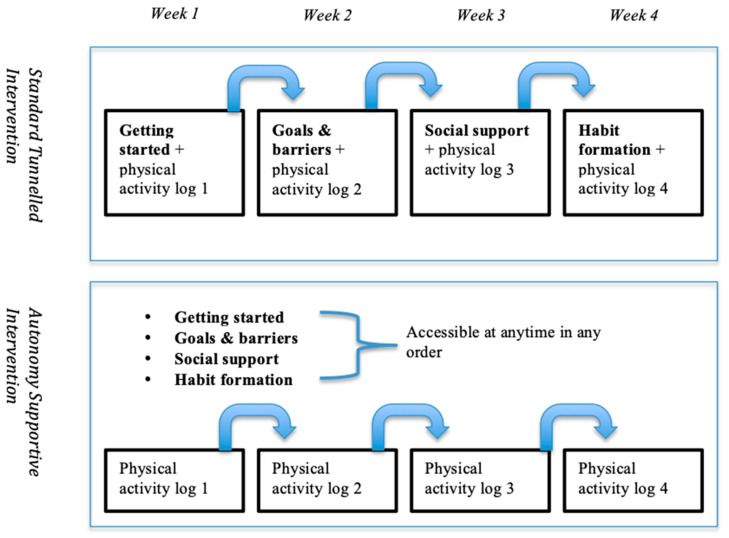
Tunneled intervention (**top**) and free-choice website intervention (**bottom**) architecture. The physical activity log (assessing activity in last week) was also available to complete at any time in the free-choice architecture condition, with a new one unlocking a week after the first one was completed.

**Figure 2 ijerph-17-07920-f002:**
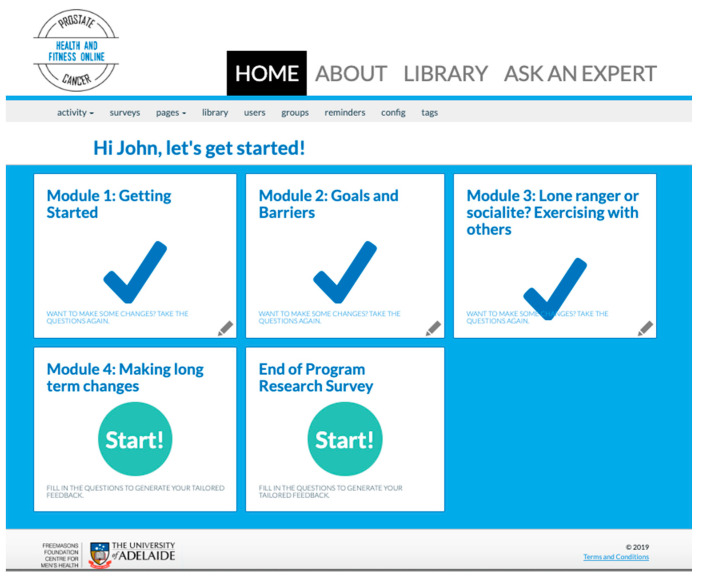
Tunnel website homepage (administrator view). A new module was unlocked each week, over four weeks; physical activity self-monitoring and feedback were provided within each module, alongside the “once-off” content designated to that week.

**Figure 3 ijerph-17-07920-f003:**
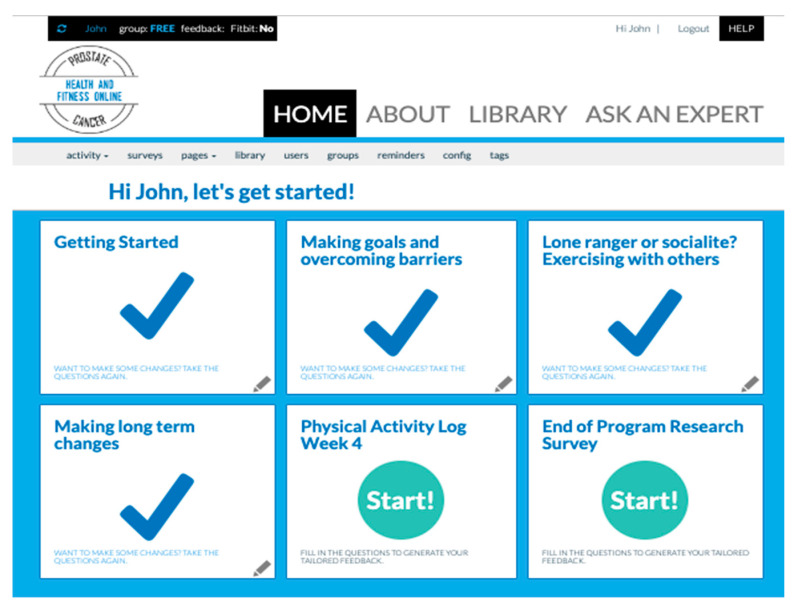
Free-choice website homepage (administrator view). This condition was designed to be autonomy supportive. Users could click on any topic at any time. Physical activity self-monitoring and feedback were supported by clicking on the physical activity log module. This automatically updated according to what week of the program the user was up to.

**Figure 4 ijerph-17-07920-f004:**
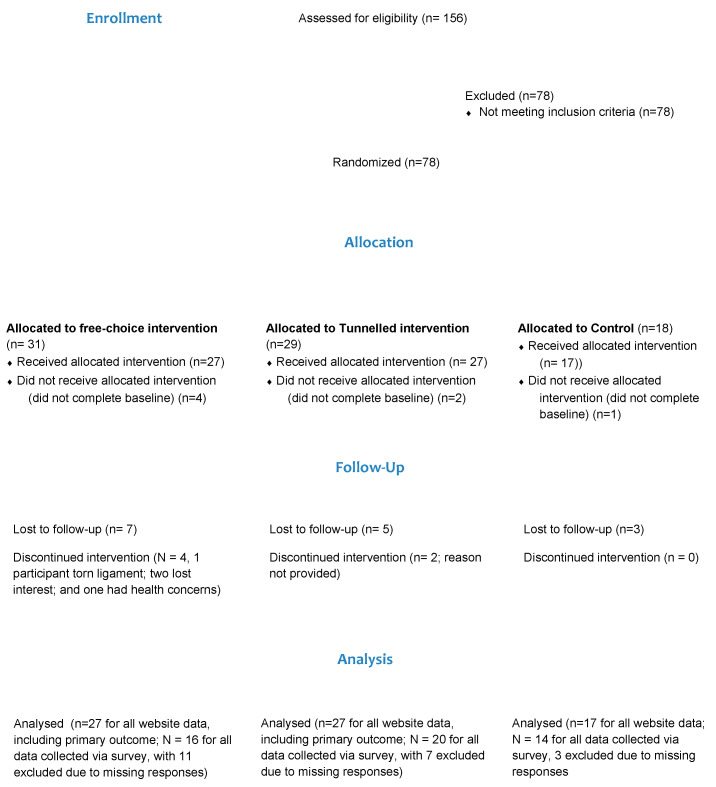
Consort flow diagram illustrating participant flow through the trial.

**Figure 5 ijerph-17-07920-f005:**
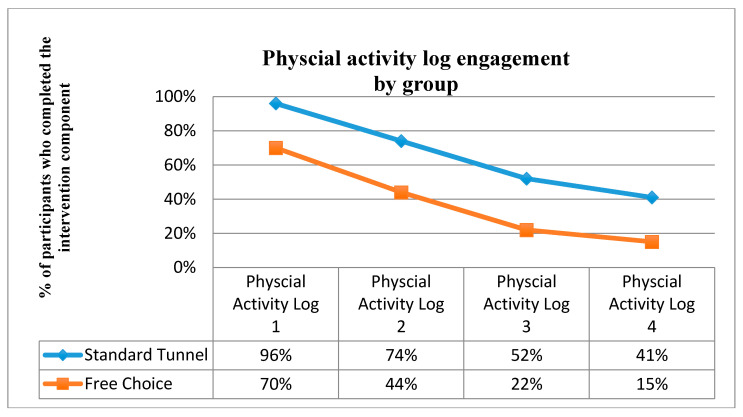
Website module engagement by group.

**Table 1 ijerph-17-07920-t001:** Participant characteristics (complete baseline data N = 71).

Characteristic		Standard Tunnel N = 27	Free Choice N = 27	Control Arm N = 17	Total N = 71
Age	Range (years)	55–91	50–79	38–79	38–91
	Mean (SD)	68.9 (9.75)	64.9 (9.27)	65.5(9.82)	66.6 (9.66)
Education	Post-grad	6 (22.2%)	4 (14.8%)	2 (11.8%)	12 (15.6%)
	University	9 (33.3%)	8 (29.6%)	9 (52.9%)	26 (33.8%)
	Trade/diploma	7 (25.9%)	7 (25.9%)	4 (23.5%)	18 (23.4%)
	High school	5 (18.5%)	8 (29.6%)	2 (11.8%)	15 (19.5%)
Marital status	Married/partner	23 (85.2%)	24 (88.9%)	13 (76.4%)	60 (84.5%)
	Separated/single	4 (14.8%)	3 (11.1%)	4 (23.6%)	11 (15.5%)
Employment	Full time	5 (18.5%)	8 (29.6%)	3 (17.6%)	16 (22.5%)
	Part time/causal	1 (3.7%)	4 (14.8%)	0 (0%)	5 (7.1%)
	Self-employed	4 (14.8%)	2 (7.4%)	1 (5.9%)	7 (9.8%)
	Retired	15 (55.6%)	12 (44.4%)	11 (64.7%)	38 (53.5%)
	Other	4 (5.6%)	1 (3.7%)	0 (0.0%)	5 (7.1%)
Location	Major city	15 (55.6%)	13 (48.1%)	7 (41.2%)	35 (45.5%)
	Inner regional	7 (25.9%)	10 (37.0%)	4 (23.5%)	21 (27.3%)
	Outer regional	4 (14.8%)	2 (7.4%)	6 (35.3%)	12 (15.6%)
	Remote	1 (3.7%)	2 (7.4%)	0 (0.0%)	3 (3.9%)
Cancer stage	Stage 1	4 (14.8%)	2 (7.4%)	3 (17.6%)	9 (11.7%)
	Stage 2	12 (44.4%)	9 (33.3%)	4 (23.5%)	25 (32.5%)
	Stage 3	10 (37.0%)	16 (59.3%)	9 (52.9%)	35 (45.5%)
	Unknown	1 (3.7%)	0 (0.0%)	1 (5.9%)	2 (2.6%)
Cancer treatment †	Surgery	25 (85%)	19 (70%)	14 (82%)	56 (79%)
	Active surveillance	1 (4%)	1 (4%)	3 (18%)	5 (7%)
	Radiotherapy	11(41%)	9 (33%)	5 (29%)	25 (35%)
	Hormone	5 (19%)	4 (15%)	3 (18%)	12 (17%)
	Other	0 (0%)	1(4%)	0 (0%)	1 (1%)
Time since treatment (years)	Range	0.08–13	0.08–10	0.17- 8	0.08–13
	Means (SD)	3.5 (3.75)	2.7 (2.6)	2.3 (2.34)	2.9 (3.0)
Time since diagnosis (years)	Range	1–16	1–12	1–19	1–19
	Mean (SD)	6.25 (4.15)	4.18 (3.4)	4.5(4.7)	5.1 (4.4)
BMI	Range	17–35	21–349	22–33	17–39
	Mean	26.3 (4.6)	28.4(3.97)	26.7 (3.17)	27.2 (0.41)
Co-morbidities (N %)	No issues	2 (7.7%)	5 (18.5%)	5 (29.4%)	12 (16.9%)
	1	9 (34.6%)	8 (29.6%)	2 (11.8%)	19 (26.8%)
	2	9 (34.6%)	8 (29.6%)	6 (35.3%)	23 (32.4%)
	3 or more	6 (22.2%)	6 (22.2%)	4 (23.4%)	17 (23.9%)

† Participants may have indicated multiple cancer treatments.

**Table 2 ijerph-17-07920-t002:** Physical activity outcomes.

MVPA, mins/Week	*Baseline* *Mean (SD)* *Range*	*Follow up* *Mean (SD)* *Range*	*Difference* *Mean (SD)* *Range*	*p*
Free choice N = 16	226.3 (252.4) 0, 900	280.62 (221.7) 0, 900	+54.37 (326.7) −600, 820	0.13 (adjusting for baseline only). 0.30 (adjusting for all covariates)
Tunnelled N = 20	232.5 (186.6) 0, 660	230.50 (199.9) 0, 900	−2 (267.5) −380, 620	
Control N = 14	292.1 (283.8) 0, 900	387.14 (213.3) 40, 900	+95 (317.6) −690, 420	
*Resistance training score*				
Free choice N = 16	20 (43.9) 0, 150	22.5 (32.2) 0, 100	+2.5 (60.5) −150, 100	0.63 (adjusting for baseline only) 0.22 (adjusting for all covariates) *
Tunnelled N = 20	11.7 (33.8) 0, 150	23.4 (47.9) 0, 210	+11.75 (SD 62.1) −150, 210	
Control N = 14	2.4 (5.7) 0, 21	19.1 (26.7) 0, 75	+16.71 (SD 28.5) −15, 12	

* Location was a significant predictor of resistance training score in the adjusted model, with living outside of a major city associated with greater resistance training than living in a major city (*b* = 26.45, 95% CI 0.57, 52.34).

**Table 3 ijerph-17-07920-t003:** Engagement, usability, relevance score comparison across the three study groups (N = 50).

*Measure* *Range*	*Free Choice* M, (SD) Range	*Tunnelled* (M, SD) Range	*Control* (M, SD) Range	*p*-Value	*Adjusted**p*-Value
E-health engagement scale (1–5)	2.3(0.83) 1, 3.5	2.4(0.87) 1.4, 4.09	2.4 (0.79) 1.3, 3.7	0.98	0.94
SUS—usability (0–100)	56.4 (12.2) 40, 82.5	67.4 (14.6) 40, 95	57.7 (17.5) 15, 85	0.06	0.08
Average perceived relevance (1–5)	2.4 (1.1) 1, 4	2.8 (1.3) 1, 4.3	2.4 (1.3) 1, 4	0.56	0.57

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
