# Peer review of "Optimising Web-Based Computer-Tailored Physical Activity Interventions for Prostate Cancer Survivors: A Randomised Controlled Trial Examining the Impact of Website Architecture on User Engagement"

_ijerph, 2020, doi:10.3390/ijerph17217920_

Round 1
Reviewer 1 Report
Quite interesting how the manuscript is a randomized process evaluation - pretty cool. I would only recommend some minor grammatical changes (e.g., removing the word "that" and consider elaborating when you begin a sentence with the word "This").

Author Response
Reviewer: Quite interesting how the manuscript is a randomized process evaluation - pretty cool. I would only recommend some minor grammatical changes (e.g., removing the word "that" and consider elaborating when you begin a sentence with the word "This").
Response: Thank you for taking the time to review our manuscript and for your positive feedback. We have reviewed the manuscript and made grammatical changes throughout to improve readability. A revised version of the manuscript is attached.
Reviewer 2 Report
This study provides valuable insights into how website architecture attempts to influence the effectiveness of computer-tailored interventions. I think it can be accepted already, but it is desirable to make the following modifications.
Page2.Line77 It may be an irrelevant point about cancer because it is not a specialty, but I would like to ask women to confirm whether breast cancer can be applied to the reference to prostate cancer.
Page11.Figure4 I think it's a careless mistake, but it's unnatural that there are no lines in Figure 4.
Page17.Line507 ”Current wearable trackers are disease non-specific” ---> What are the characteristics of wearable trackers that specialize in cancer? If possible, it would be easy if you could understand the characteristics of wearable trackers that specialize in prostate cancer.
Reference The titles of papers are not unified, such as all or abbreviated.
Conflicts of Interest Please make a final check to make sure that there is no relationship between the authors and PCHF that corresponds to COI.
Finally, it's just an impression. I think that the research of Computer-tailord has entered into in-depth research. However, I may not understand it, but I feel that the tailoring of this research is a little too simple. In addition to academic aspects, I would like you to make efforts in the future to achieve clinical results such as steadily increasing physical activity.
Author Response
Reviewer comment: This study provides valuable insights into how website architecture attempts to influence the effectiveness of computer-tailored interventions. I think it can be accepted already, but it is desirable to make the following modifications.
Response: Thank you for taking the time to review our manuscript, and for your positive feedback and suggestions for improvement. Please see our responses to your specific comments below. The revised manuscript is also attached.
Reviewer comment: Page2.Line77 It may be an irrelevant point about cancer because it is not a specialty, but I would like to ask women to confirm whether breast cancer can be applied to the reference to prostate cancer.
Response: This paragraph (page 2, line 76-94) aims to describe the limited research that has been conducted on website architecture and computer-tailoring to date. While the breast cancer study is similar in that it focuses on cancer and physical activity, it investigates the timing of three different tunnelled delivery structures – rather than a tunnelled structure versus a more autonomous structure. In this sense, even if the breast cancer study results were generalisable to prostate cancer, the study still does not investigate the impact of an autonomous structure versus a tunnelled structure. The key point we make about this literature, in combination with the few non-cancer specific studies conducted is that website architecture does seem to influence engagement, but no studies have focused on prostate cancer (see line 74), or on the impact of providing a completely autonomous structure compared to a tunnelled structure.
To make our key point clearer we have made two minor adjustments to the manuscript. This includes an explicit statement that the impact of website architecture has not been explored among prostate cancer survivors (Line 74), and that our aim is to assess the impact of an autonomy supportive intervention among prostate cancer survivors specifically (Line 98).
Reviewer comment: Page11.Figure4 I think it's a careless mistake, but it's unnatural that there are no lines in Figure 4.
Response: Figure 4 follows the consort template. There may have been an issue with upload. We have made some minor adjustments and re uploaded.
Reviewer comment: Page17.Line507 ”Current wearable trackers are disease non-specific” ---> What are the characteristics of wearable trackers that specialize in cancer? If possible, it would be easy if you could understand the characteristics of wearable trackers that specialize in prostate cancer.
In this paragraph we suggest that wearables that log exercise and provide behaviour change support may need to be adapted to better suit cancer patients, or be paired with computer tailoring. To highlight what we mean we have included the following in text:
“For example, step count goals that are adjusted based on treatment-related side-effects and behaviour change support that acknowledges unique barriers relating to cancer symptoms (e.g., incontinence, fatigue).”
Reviewer comment: Reference The titles of papers are not unified, such as all or abbreviated.
Response: Thank you for noting this. We have now updated our references using the MDI endnote style guide.
Reviewer comment: Conflicts of Interest Please make a final check to make sure that there is no relationship between the authors and PCHF that corresponds to COI.
Response: We can confirm that we have no conflict of interest to declare.
Reviewer comment: Finally, it's just an impression. I think that the research of Computer-tailord has entered into in-depth research. However, I may not understand it, but I feel that the tailoring of this research is a little too simple. In addition to academic aspects, I would like you to make efforts in the future to achieve clinical results such as steadily increasing physical activity.
Response: We agree with this comment. Of note, the research team have more clinically focused computer-tailoring studies underway where the computer-tailored content is much more comprehensive.
The aim of the study described in this manuscript was to inform website design decisions for our clinical interventions. We do acknowledge the limitations of the tailoring used and that more comprehensive tailoring would provide greater insights (line 521-545).
E.g., “ In order to be able to observe any added benefits of self-selecting tailored modules rather than being guided through them, these limitations in the tailored content need to be addressed. “

Reviewer 3 Report
Dear Authors:
Thank you for submitting the manuscript entitled "Optimising web-based computer-tailored physical activity interventions for prostate cancer survivors: A randomised controlled trial examining the impact of website architecture on user engagement". It is an interesting topic. The reviewer appreciates the authors' attempt to assess the influences of website architecture on patients' behavioral engagement. I have the following detailed comments.
1. Introduction
- Background and controversy are adequately introduced.
- The purpose is appropriate.
2. Methods
- How is the duration of the test determined? It seems the number of participants is below the calculated sample size, would this affect the outcome of the study?
- Line 171: This paragraph is repeating the table content. I would suggest to remove it as it makes the whole method section long and redundant.
- Line 240: When were the participants asked about their preference?
- Primary outcomes are appropriate.
3. Results
- Figure 4 seems like missing some lines in the flowchart.
- Again, the actual number of participants is less than the calculated sample size. The only significance between groups was accessing the social support and habit formation modules. Have you considered elongating the study time?
4. Discussion
- Line 556: Please justify why a longer duration of study would result in a higher retention rate?
- Please compare with previous studies and discuss what is new and unique in this study.
5. Conclusions
- The conclusion is OK.
6: Reference
- The reference is ok.
Author Response
Dear Authors:
Thank you for submitting the manuscript entitled "Optimising web-based computer-tailored physical activity interventions for prostate cancer survivors: A randomised controlled trial examining the impact of website architecture on user engagement". It is an interesting topic. The reviewer appreciates the authors' attempt to assess the influences of website architecture on patients' behavioral engagement. I have the following detailed comments.
Response: Thank you for taking the time to review our manuscript. Please see our response and description of changes below.
- Introduction
- Background and controversy are adequately introduced.
- The purpose is appropriate.
Response: Thank you. No changes requested.
- Methods
- How is the duration of the test determined? It seems the number of participants is below the calculated sample size would this affect the outcome of the study?
Response: A 4-week intervention period was chosen as previous studies have demonstrated that number of logins and module usage typically declines significantly within the first 1-4 weeks. We have added the following to the procedure section to make this clear.
“This Intervention time frame was considered adequate given other web-based modular intervention studies have demonstrated significant non usage attrition within one to four weeks [23,31].”
As is typical, the recruitment period was based on the time and resources available to the research team. We have already addressed our low study numbers in the discussion. E.g.,
Line 564 -Further, even though the recruitment period was doubled in time, the study did not reach the recruitment target, similar to issues noted in other men’s health research [54,55]. This indicates a major feasibility issue, and limits the ability to discern differences, particularly in secondary outcomes.
Line 580 - Despite not meeting the recruitment target, this study was still able to identify significant differences in the main outcome. Future prostate cancer studies could utilise additional recruitment avenues, such as the use of cancer registries to obtain more representative samples.
- Line 171: This paragraph is repeating the table content. I would suggest to remove it as it makes the whole method section long and redundant.
Response: Thank you for this suggestion. However, as the table is only included as an Appendix, and we think it is important that readers are aware of what the once off modules contained in order to interpret the results, we have decided to retain it. If the editor or yourself feel strongly about the removal of this paragraph we are happy to reconsider.
- Line 240: When were the participants asked about their preference?
Response: This was asked as part of the baseline survey. We agree this was unclear and have included the following sentence “Intervention architecture preference was also collected via the self-report survey at baseline” (Line 243)”
- Primary outcomes are appropriate.
Response: Thank you. No changes requested.
- Results
- Figure 4 seems like missing some lines in the flowchart.
Response: This was also noted by reviewer 2. However, Figure 4 follows the consort template. There may have been an issue with upload. We have made some minor adjustments and re uploaded. Please see attached.
- Again, the actual number of participants is less than the calculated sample size. The only significance between groups was accessing the social support and habit formation modules. Have you considered elongating the study time?
Response: While the number of participants is less than our sample size calculation, we were able to see significant differences for the main study outcome – The number of physical activity logs that were completed.
On top of this, we were also able to detect differences in in the proportion of users accessing the social support and habit formation modules (in favour of the free choice group).
Unfortunately, we are not able to elongate the study time. We have however noted several methodological aspects in the discussion that need to be considered in the future in order to be able to explore the intersect between usage and behavioural and clinical outcomes. This includes addressing noted limitations with the intervention content (line 533-547), using stricter inclusion criteria and objective measures of behaviour ( which would reduce ceiling effects and the large standard deviations associated with self-reported physical activity data – both of which would improve power; Line 580-585), and using additional recruitment strategies (such as cancer registries; Line 591-593).
- Discussion
- Line 556: Please justify why a longer duration of study would result in a higher retention rate?
Response: This sentence states that the shorter duration of our study should result in higher retention than studies that are longer in duration. We have revised this sentence to improve clarity/ avoid confusion. This sentence now reads
“Given the relatively short duration of the current study, and that drop-out tends to increase overtime, a higher retention rate could be expected in our study.”
- Please compare with previous studies and discuss what is new and unique in this study.
Response: Thank you for this suggestion. We feel that our findings have been comprehensively discussed in the context of the limited research available, including:
- Available literature on website intervention architecture (line 493-513)
- Literature on non-usage of physical activity logs (our primary outcome), strategies to address this, and possible future directions (line 514-532)
- Relevance levels achieved in computer tailored interventions and factors that need to be addressed to improve perceived relevance (line 533-557).
However, we agree that what this study adds to this literature could be made more clear. As such, we have added the following to the conclusion:
“This study provides further support that a tunnelled approach may be best suited to encourage self-monitoring of physical activity beahviour, however there are important trade-offs to consider that may impact on intervention efficacy among men with prostate cancer”
- Conclusions
- The conclusion is OK.
Response: Thank you.
6: Reference
- The reference is ok.
Response: Thank you.

Round 2
Reviewer 3 Report
Dear Authors:
Thank you for submitting the revised manuscript.
My concerns and comments about the duration of the test and the number of participants have been addressed.
The Method now provides more details of the study design.
Figure 4 has been updated.
Discussion about the unique aspects of the current study has been added.
Comments from other reviewers have also been addressed.
Nice job!